# Investigation of Gas-Water-Sand Fluid Resistivity Property as Potential Application for Marine Gas Hydrate Production

**DOI:** 10.3390/e24050624

**Published:** 2022-04-29

**Authors:** Zhiwen Zhang, Xiaosen Li, Zhaoyang Chen, Yu Zhang, Hao Peng

**Affiliations:** 1Key Laboratory of Gas Hydrate, Guangzhou Institute of Energy Conversion, Chinese Academy of Sciences, Guangzhou 510640, China; zhangzw@ms.giec.ac.cn (Z.Z.); lixs@ms.giec.ac.cn (X.L.); zhangyu1@ms.giec.ac.cn (Y.Z.); penghao@ms.giec.ac.cn (H.P.); 2Guangdong Provincial Key Laboratory of New and Renewable Energy Research and Development, Guangzhou 510640, China; 3University of Chinese Academy of Sciences, Beijing 100027, China; 4State Key Laboratory of Natural Gas Hydrates, CNOOC Research Institute, Beijing 100028, China

**Keywords:** natural gas hydrates, sand production, multiphase fluid, phase fraction measurement, electrical resistivity, combined resistivity method

## Abstract

The phase fraction measurement of gas-water-sand fluid in downhole is an important premise for safe and stable exploitation of natural gas hydrates, but the existing phase fraction measurement device for oil and natural gas exploitation can’t be directly applied to hydrate exploitation. In this work, the electrical resistivity properties of different gas-water-sand fluid were experimentally investigated using the multiphase flow loop and static solution experiments. The effect of gas phase fraction and gas bubbles distribution, sand fraction and sand particle size on the relative resistivity of the multiphase fluid were systematically studied. The measurement devices and operating parameters were also optimized. A novel combined resistivity method was developed, which demonstrated a good effect for the measurement of phase fractions of gas-water-sand fluid, and will have a good application potential in marine natural gas hydrates exploitation.

## 1. Introduction

Natural gas hydrates is a solid compound formed by water and small molecule gas in a low temperature and high pressure environment [1]. Natural gas hydrates (mainly methane hydrate) is an efficient and clean energy source with great application prospects [2]. Natural gas hydrates exploitation methods currently include depressurization [3,4], thermal stimulation [5,6,7], inhibitor injection [8], replacement with CO_2_ [9,10], solid-state fluidization [11] and in-situ catalytic oxidation [12]. However, no matter which method is adopted for exploitation, it is a necessary link for hydrate exploitation to collect and transport the gas-water mixture produced by hydrate decomposition in hydrate reservoir to the offshore platform through exploitation wellbore and riser. Because most of the marine gas hydrates exist in the shallow argillaceous silt sediments on the seabed, it is very easy to produce sand in the mining process [13]. Excessive sand production will cause shaft blockage, production equipment damage and other mining accidents [14]. Although excessive sand control strategies, such as injecting a large amount of adhesive or setting a small screen aperture, can effectively prevent the sand production, the gas-water mixture generated by hydrate decomposition is also difficult to move in the reservoir or enter the wellbore from the reservoir. This will lead to reduced gas production. In the actual production process, there will always be sand in the wellbore. At the same time, the gas and water in the exploitation wellbore and riser are closely mixed, and the temperature and pressure of the system are very easy to enter the area where the hydrate is stably generated, resulting in a large number of secondary hydrate generation and safety accidents such as blockage of the wellbore, riser and production equipment [15]. Therefore, in the process of natural gas hydrates production, the gas-liquid-solid (sand + hydrate) three-phase fluid in the exploitation wellbore and riser has been in dynamic change, with great randomness and burst. Therefore, it is of great practical engineering significance to study the in-line measurement technology to measure the phase fraction of gas-water-sand fluid in hydrate exploitation wellbore. Adjusting the exploitation strategy and related exploitation parameters according to the real-time sand production is beneficial to the long-term stable operation of the exploitation system.

Multiphase fluid widely exists in a variety of scenarios, including oil and natural gas extraction, fluidized beds, and sandy rivers. Researchers have developed a variety of phase fraction measurement methods based on different principles, including optical methods, acoustic methods, mechanical methods, electrical methods, thermal methods, etc. (1) The non-contact optical methods use gamma rays, X-rays, microwaves, etc., and measure phase fraction according to the different attenuation rates of rays in different media [16]. Gamma-ray and microwave methods are widely used in the oil and natural gas extraction, but the presence of radiation requires special training for operators. The fiber probe is a contact optical method, which uses the different refractive indices of light in each phase of the multiphase fluid to identify the local phase state. With the development of image processing technology, the method of using high-speed cameras to collect multiphase fluid images and then processing them to obtain phase fraction has also been proposed [17]. (2) Acoustic methods include active ultrasound and passive ultrasound, both of which use the different attenuation characteristics of acoustic waves in different media to measure the phase fraction of multiphase fluid [18,19,20]. (3) The mechanical method uses electromagnetic drive to vibrate local pipe sections, and reflects the phase fraction of multiphase fluid through vibration characteristics. (4) Electrical methods include the impedance method using the different electrical properties of the medium itself and the erosion (ER) method using metal sheet erosion to measure the solid phase fraction [21,22] and so on, as shown in Table 1. The impedance method specifically includes the wire mesh sensor method [23,24,25,26], the electrical tomography method [27,28,29,30,31], the conductance probe method [32], the capacitively coupled contactless conductivity detection method [33] and so on. The electrical method has low cost, no radiation, and simple structure [34,35], and is suitable for hydrate exploitation wellbore with high pressure and variable salinity. Therefore, the electrical method for measuring phase fraction has great application prospects in the measurement of phase fraction in hydrate exploitation wellbore. Since the liquid phase of the multiphase fluid in the hydrate exploitation wellbore is seawater, and the sand production process is generally accompanied by a large amount of water production, the continuous phase of the gas-water-sand fluid is the conductive phase. Therefore, it is suitable to use the conductivity method instead of the capacitance method to measure the phase fraction of gas-water-sand fluid.

Among various electrical phase fraction measurement methods, the contacting electrical method is simple and practical, and measures the fluid impedance information through several pairs of electrodes in direct contact with the fluid. According to the different electrode structures, it can be divided into ring type, opposite-wall type and parallel type. Andreussi et al. [36] proposed an electrode arrangement method including three ring electrodes, and developed the theoretical basis of ring electrodes. Kytomaa et al. introduced a guard electrode to support two opposite-wall electrodes, so that the electrode sensitive field will be closer to the cross-sectional slice of the pipe. Coney et al. [37] studied the theoretical basis of parallel rectangular electrodes. They used parallel rectangular electrodes to measure the liquid film thickness of a separated flow (Annular or laminar flow) and obtained a relationship between the liquid film thickness and the equivalent conductance of two-phase fluid. In order to obtain more impedance information and avoid the influence caused by the uneven spatial distribution of phase fraction, the method of combined electrodes and rotating excitation are often used in the contacting electrical phase fraction measurement process. Merilo et al. [38] designed a pipeline fluid conductance sensor with a six-electrodes structure, which formed a compensated rotating electric field inside the pipeline under test by using three pairs of sequentially excited electrodes. Tournaire et al. [39] compared the conductance sensor with the six-electrodes structure with the traditional two-electrodes structure conductance measurement sensor, and proved that the six-electrodes conductance sensor has better phase fraction measurement performance. Liu et al. [40] measured the pure liquid-phase conductivity of multiphase fluid by covering the electrodes with screens. Qing et al. [41] measured the phase fraction of TBAB hydrate slurry using a combined electrodes of a micro electrode and two ring electrodes. Wang et al. [42,43,44,45] used the combined method of the opposite-wall electrodes and the side-wall electrodes to realize the first purely electrical measurement of the phase fraction of the two-phase fluid under the changing salinity.

In marine natural gas exploitation, the produced fluid is a gas-water-sand mixture with large gas and water phase fraction; in conventional oil exploitation, the produced fluid is mainly an oil-water mixture with large oil phase fraction; in conventional natural gas exploitation, the produced fluid is mainly a gas-water mixture with large gas phase fraction. The difference in the tested systems makes it difficult for the current multiphase fluid testing technology applied to conventional oil and natural gas systems to be directly applied to the downhole multiphase flow monitoring of marine natural gas exploitation. Therefore, there is an urgent need to study the influence mechanism and change rule of its resistivity properties and develop corresponding measurement technologies and methods according to the multiphase fluid properties s of marine natural gas exploitation.

The actual wellbore pressure of hydrate exploitation is very high and below the sea surface, resulting in the high cost of downhole research. Therefore, in this work, we conduct simulation research in the laboratory to provide a theoretical basis for practical wellbore application. A multiphase flow loop and a measurement setup for static mixture were built in the laboratory. The electrical resistivity properties of the gas-water-sand fluid were experimentally studied. A combined resistivity method was proposed to measure the phase fraction of gas-water-sand fluid in wellbore during exploitation of natural gas hydrates. The operating parameters and the hardware design of the measurement devices, such as the excitation frequency and the pipe wall material, were also studied and optimized in this work.

## 2. Experimental Section

### 2.1. Measurement Principle of the Combined Resistivity Method

For two-phase fluid, such as gas-liquid fluid, the phase fraction determines the relative resistivity of the fluid, so the phase fraction can be obtained by the relative resistivity of the fluid. However, the relative resistivity of gas-water-sand fluid in hydrate production is determined by both gas phase fraction and sand phase fraction. Therefore, it is necessary to separate the gas-water-sand fluid, and then measure the relative resistivity of gas-water-sand fluid and water-sand fluid respectively, so as to realize the simultaneous measurement of gas phase fraction and sand phase fraction. In this work, the relative resistivity of each component is measured by the combined resistivity method.

The principle of the combined resistivity method proposed in this work is shown in Figure 1, where the red arrows represent the electric field lines. The combined resistivity method adopts three kinds of electrodes, and the functions of the three electrodes are introduced as follows: (1) The opposite-wall electrodes are installed on the side of the pipe, facing each other. Its sensitive field covers almost the entire pipe cross section, so the total resistivity ρ_G+L+S_ of the gas-water-sand fluid can be measured. (2) The side-wall electrode refers to electrode that is installed on the sidewall of the pipe and the distance between electrode pairs is only a few millimeters. The sensitive field of the side-wall electrode is relatively small, generally only covering a distance of a few millimeters from the wall surface. The resistivity ρ_L+S_ of the water-sand fluid can be measured by the side-wall electrode. (3) The micro electrodes refer to electrode pairs with extremely small electrode spacing, and it is difficult for sand particles to enter the area between the electrode pairs, so that the resistivity ρ_L_ of pure solution (without sand particles) can be measured.

### 2.2. Materials

In this work, the deionized water (Produced by Efund Water Purifier, Nanjing, China), sodium chloride (99.5% pure, Macklin, Shanghai, China), magnesium sulfate anhydrous (99.99% pure, Macklin, Shanghai, China) were used to prepared the brine water. The quartz sand (Aladdin, Shanghai, China) with particle sizes of about 2 μm, 5 μm, 10 μm and 60 μm were used as sand produced from gas hydrate formation. The particle size distribution of these quartz sand was measured by Mastersizer 2000E (Kidlington, UK), and are shown in Figure 2.

### 2.3. Experimental Apparatus

#### 2.3.1. Multiphase Flow Loop Setup

In order to study the resistivity phase fraction measurement of gas-water-sand fluid in hydrate exploitation wellbore, an experimental loop device for electrical resistivity properties of multiphase flow was built in this work. The schematic diagram and device photos of the multiphase flow loop are shown in Figure 3 and Figure 4, respectively. The multiphase flow loop setup is composed of circulating loop and resistance measuring device.

The circulating loop includes temperature control unit, flow control unit and gas and sand injection unit. (1) The temperature control unit: The temperature of the flowing fluid in the Loop is measured by a K-type thermocouple with an accuracy of 0.1 °C, and controlled by a constant-temperature circulation water bath (Xutemp, Hangzhou, China) with a temperature range of −25 °C to 90 °C and a temperature fluctuation of ±1 °C. (2) The flow control unit: A stainless steel corrosion-resistant self-priming water pump (Minglei, Nantong, China) with a maximum flow rate of 6 m^3^/h provides power for the loop. A frequency converter with frequency range of 0 to 50 Hz is used to control the water pump flow. The flow of water is measured by the liquid rotameter (LZT-1005G, Taizhou, China) with a range of 10–70 L/min. (3) The gas and sand injection unit: An air compressor supplies gas to the loop. The flow of injected gas is measured by the gas rotameter (LZB-6, Xiangjin, Ningbo, China) with a range of 0–600 L/h. A certain amount of quartz sand was weighed out by a balance (the range is 3 kg, the accuracy is 0.01 g, Anheng, Shenzhen, China) and added it into the mixing tank with the stirring speed range of 0–130 r/min.

The resistance measuring device is composed of test unit and resistance measurement circuit. The test unit is composed of measuring pipe section and electrodes, which are described in detail below.

The measuring pipe section: The structure of the test pipe section is shown in Figure 5, which consists of double-layer pipes with a total length of 900 mm. The outer pipe is made of plexiglass, with an outer diameter of 70 mm and an inner diameter of 60 mm. There are 4 circular holes facing each other every 100 mm, and the diameter of the circular holes is 10 mm. The inner pipe material (plexiglass or stainless steel) can be selected according to the experimental requirements. Its outer diameter is 58 mm (slightly smaller than the inner diameter of the outer pipe for easy installation), its inner diameter is 50 mm, and its length is 300 mm. On the inner pipe, punch 4 circular holes facing each other every 100 mm, and the diameter of the circular holes is 12 mm. The electrode, temperature sensor and pressure sensor were installed on the pipe wall through the circular hole, and the installation position of the electrode can be easily changed.

The electrodes: Three types of electrodes were used in this work which are shown in Figure 6, and the details are as follows. (a) The opposite-wall electrode adopts a circular glassy carbon electrode with a diameter of 4 mm, which is covered with insulating material and has a total diameter of 10 mm. (b) The upper and lower electrodes of the micro electrode are made of stainless steel, and the middle is filled with insulating materials. The distance between the two electrodes is 5 μm in this work, which is equal to the thickness of the insulating material. (c) The side-wall electrode is made of stainless steel, and its outside is covered with insulating materials. The diameter of the electrodes was 0.5 mm, the distance between the electrodes was 2 mm, and the outer diameter was 10 mm. In the research of Wang et al. [45], an insulating rubber rod was used inside the pipeline to determine the sensitive field range of the sidewall electrodes. The results show that the sensitive field range of the sidewall electrodes with a spacing of 2 mm is about 5 mm.

The resistance measurement circuit: In this work, multiple pairs of electrodes were used for measurement, and the multiple pairs of electrodes adopt the cyclic excitation method, that is, only one pair of electrodes is excited at each time point, so as to avoid electric field interference between the electrode pairs. We used a custom circuit board to cyclically switch the excitation signal, which uses a relay controlled by a microcontroller to control the switching. The circuit board has a total of four output channels, which can realize the cyclic excitation of four pairs of electrodes. In this work, a signal generator (RIGOL LG1022, Suzhou, China) was used to generate AC signal, which can output various signals such as sine wave and square wave with a frequency range of 0–25 MHz and a voltage amplitude range of 0–20 V. Agilent 34970A was used to collect the voltage signal and current signal at both ends of the electrode pair in real time. The connection mode of the measurement circuit is shown in Figure 7.

#### 2.3.2. Measurement Setup for Static Mixture

A static experimental device was also built in this work which is shown in Figure 8. The main body is a cylinder, and the inner diameter of the cylinder is 50 mm, which is the same as the inner diameter of the loop mentioned above. The cylinder was placed in a large double-layer beaker to ensure a constant temperature. A pair of opposite-wall electrodes were installed at a distance of 100 mm from the bottom of the cylinder. Some horizontal lines were marked every 2 cm above the electrodes. A magnetic stirrer was used to ensure the uniformity of sand particle distribution.

### 2.4. Experimental Procedures and Methods

#### 2.4.1. Loop Experiment Measurement

After the setup was rinsed and tested, some brine water with a given salinity was added into the multiphase flow loop setup, and some quartz sand were added into the mixing tank under a stirring speed of 100 r/min. Then the water pump and the constant-temperature circulation water bath were turn on. The inverter was set at 25 Hz frequency (remain unchanged in this article) so that the flow of the liquid remains 35 L/min (measured by rotameter). The output waveform of the signal generator was set as sine wave, and its signal frequency and amplitude were set according to the experimental requirements. After temperature of the water-sand fluid reached the predetermined temperature, the gas was bubbled into the loop from the bottom of the test pipe through the gas rotameter and gas distributor by air compressor. The electrical signal data of all pairs of electrodes are acquired by the Agilent 34970A at an acquisition frequency of 1 Hz, and recorded by a computer automatically.

#### 2.4.2. Static Experiment Measurement

The static experiments of different simulated seawater-sand systems were conducted on the static device. The experimental methods and operating parameters were the same as those in the loop experiment except the mixed fluid was not flowing.

#### 2.4.3. Data Processing Method

(1)Calculation method of relative resistivity

In this work, relative resistivity refers to the ratio of the ratio of the total resistivity of the multiphase fluid to the resistivity of the composition phases. The relative resistivity was adopted to character the effect of gas and sand addition on the electrical resistivity properties of the multiphase flow, which was calculated by the following formulas:(1)$$\rho_{r,G}=\frac{\rho_{G+L+S}}{\rho_{L+S}}=\frac{R_{G+L+S}/A_{1}}{R_{L+S}/A_{2}}=\frac{R_{G+L+S}}{R_{L+S}}\frac{A_{2}}{A_{1}}$$
(2)$$\rho_{r,S}=\frac{\rho_{S+L}}{\rho_{L}}=\frac{R_{S+L}/A_{2}}{R_{L}/A_{3}}=\frac{R_{S+L}}{R_{L}}\frac{A_{3}}{A_{2}}\;$$
where ρ_r,G_ and ρ_r,S_ are the relative resistivity of gas and sand, respectively; ρ_G+L+S_, ρ_S+L_ and ρ_L_ are the resistivity of gas-water-sand fluid, water-sand fluid, and pure solution (without sand), respectively; A_1_, A_2_, and A_3_ are the electrode constants of the opposite-wall electrode, the side-wall electrode, and the micro electrode, respectively. A_2_/A_1_ and A_3_/A_2_ are obtained by measuring the resistance of the same solution. In this work, the average relative resistivity refers to the average of all relative resistivities in the acquisition time.

(2)Calibration method of phase fraction

In this work, the phase fraction refers to the ratio of the volume of each phase to the total.
(3)$$\alpha_{S}=\frac{V_{S}}{V_{G}{+V}_{L}{+V}_{S}}\;$$
(4)$$\alpha_{G}=\frac{V_{S}}{V_{G}{+V}_{L}{+V}_{S}}\;$$
where α_S_, α_G_ are the sand phase fraction and the gas phase fraction, respectively; V_G_, V_L_ and V_S_ are the volume of gas, water and sand, respectively.

For gas-water mixed fluid, the gas fraction is calculated by liquid flow and gas flow, which is used as the calibration value of gas fraction.
(5)$$\alpha_{G}=\frac{V_{G}}{V_{L}{+V}_{G}}=\frac{Q_{G}}{Q_{G}{+Q}_{L}}$$
where Q_L_ and Q_G_ represent liquid flow and gas flow, respectively.

For water-sand mixed fluid, the sand phase fraction is calculated through the volume of liquid in the system and the volume of sand, which is used as the calibration value of sand phase fraction. The volume of sand is calculated by the mass and density of sand.
(6)$$\alpha_{S}=\frac{V_{S}}{V_{L}{+V}_{S}}=\frac{m_{S}/\rho_{S}}{V_{L}{+m}_{S}/\rho_{S}}$$
where m_S_ and ρ_S_ represents the mass and density of sand, respectively.

(3)Method for calculating phase fraction by resistivity

Gas phase fraction and sand phase fraction are related to the relative resistivity of gas-water fluid and water-sand fluid respectively. The empirical relationship between them needs to be obtained through experiments. According to the measured relative resistivity data, combined with the empirical relationship, the gas phase fraction and sand phase fraction can be calculated. The calculation formula is as follows:(7)$$\alpha_{G}{=f}_{G}\left(\rho_{r,G}\right)=\frac{A_{2}}{A_{1}}\times f_{G}\left(\frac{R_{G+L+S}}{R_{L+S}}\right)\;$$
(8)$$\alpha_{S}{=f}_{S}\left(\rho_{r,S}\right)=\frac{A_{3}}{A_{2}}\times f_{S}\left(\frac{R_{S+L}}{R_{L}}\right)\;$$
where f_G_ and f_S_ represent the empirical relationship of gas phase fraction and sand phase fraction respectively.

## 3. Results and Discussion

### 3.1. Optimal Design of Experimental Device

#### 3.1.1. Selection of Excitation Frequency

The polarization effect of the electrode (redox reaction and ions directional movement caused by current) would seriously affect the measurement of resistance if direct current (DC) was used as the excitation power supply, so alternating current (AC) was used for the excitation power supply in this work. In order to optimize the excitation frequency of AC, the flow loop experiments were carried out at an excitation frequency of 100 Hz, 500 Hz, 1 kHz, 10 kHz, 50 kHz and 100 kHz, respectively. The gas-water fluid contains 1.3% NaCl, and the gas was injected by air stone. The resistance measured at different time are displayed in Figure 9 and Figure 10. It can be seen that, when the excitation signal frequency is lower than 1 kHz, the resistance fluctuates greatly with time, and the standard deviation of the resistance measured within 180 s is higher than 1835; when the excitation signal frequency decreased to 100 Hz, the resistance value became more divergent, and the standard deviation reached 12.366, which may be caused by the polarization effect of the electrode at a low excitation signal frequency. At 10 kHz excitation signal frequency, the resistance value changed very little, and exhibited an optimum measuring performance. When the excitation signal frequency increased more to 50 kHz and 100 kHz, the standard deviation of the resistance measured increased instead due to the current leakage of electrical capacity. So, the excitation frequency was fixed at 10 kHz in all the following studies.

#### 3.1.2. Effect of the Pipelines Wall Material

In the actual exploitation of s, the pipelines used are made of metal materials with small resistivity. In order to explore the effect of the pipelines wall materials on resistivity measurement, we made a comparative study of stainless steel and plexiglass inner pipe on the average relative resistivity measured by the opposite-wall electrodes in the flow loop. The gas-water fluid contains 1.3% NaCl, and the gas was injected by air stone. The repeating experimental results are displayed in Figure 11. The average relative resistivity of the fluid in the plexiglass pipe increases with the increase of the gas phase fraction, and exhibits very good linear relationship and good repetitiveness for the three experiments. However, the average relative resistivity of the flow in the steel pipe fluctuates and has not a definitive relationship with the gas phase fraction. The results show that the electrical resistivity of the pipelines wall material has a great influence on the electrical resistance measurement. The steel pipes used in the hydrate exploitation wellbore will strongly disturb the electrical field due to its small electrical resistivity. So, some measures need to be taken to avoid the influence of the wall leakage on the electrical measurement, such as insulating the pipelines wall near the measurement pipe section. The plexiglass pipe presented a good possibility for gas phase fraction measurement by electrical method, and was used in all the following studies.

#### 3.1.3. Effect of the Electrode Shape

Electrode is an important component in electrical resistivity measurement. Firstly, the electrode should be highly sensitive to the change of phase fraction in multiphase fluids. Secondly, the shape of the electrode should have a high mechanical strength and good corrosion resistance, and be easy to produce and install. In order to study the effect of electrode shape on the sensitivity of electrode to gas phase fraction in gas-water fluid, we selected two types of electrodes: circular opposite-wall electrodes and rectangular opposite-wall electrodes, and measured the resistance of gas-water fluid with different gas phase fraction in the flow loop. The gas-water fluid contains 1.3% NaCl, and the gas was dispersed by air stone. The results are displayed in Figure 12.

It can be seen from the Figure 11, for both type electrode shapes, there is a good linear relationship between the average relative resistivity and the gas phase fraction of gas-water fluid, and there isn’t obvious difference between the two electrode shapes, which indicates that the electrode shape may be designed freely according to the convenience of production and installation in the later design of resistivity measuring device for the downhole application.

#### 3.1.4. Axial Sensitive Field Range of the Opposite-Wall Electrodes

The sensitive field range of the opposite-wall electrodes affects the length of the insulation section of the downhole resistivity phase fraction measuring device, and determines whether there will be an impact between various electrodes used in the measuring device. In order to study the axial sensitive field range of the opposite-wall electrodes, a certain amount of brine water was added into the cylinder of the measurement setup for static mixture until the water level reached the horizontal line mark 2 cm away from the upper end of the electrode. After the temperature of the brine water was stabilized at 20 °C by the water bath, the signal generator and Agilent 34970A were turn on, and the voltage and current signals were collected for 5 min. After the acquisition, adding more brine water into the cylinder until the water level reaches the next horizontal line mark, and repeat the above operation. The experimental results are shown in Figure 13. As the vertical distance between the water level to electrode increases from 0 to 2 cm, the resistance of the opposite wall electrode decreases from 24.5 Ω to 24.1 Ω due to the conductive area increase. When the vertical distance increases from 2 cm to 18 cm, the resistance values measured by the electrode pair has almost no change. This indicates that the electrical field space range was limited about 4 cm away from the upper and lower ends of the electrode pair under these experimental conditions. And the sensitive field of the electrode is concentrated on a flat cross-section with little axial extension.

### 3.2. Resistivity Properties of Multiphase Fluid

#### 3.2.1. Resistivity Properties of Gas-Water Fluid

(1)Effect of gas phase fraction on resistivity properties

In the actual exploitation of natural hydrate, the injection of hot water and the addition of inhibitors will lead to the variation of pipeline fluid salinity. The change of salinity of pipeline fluid will have a great effect on the resistance of pipeline flow. So, we studied the effect of gas phase fraction on resistivity characteristics of gas-water fluid under the change of salinity. In this work, the salinity change of the fluid in the wellbore was simulated by adding 1.3%, 2.7% and 4% NaCl solutions into the flow loop.

It can be seen from Figure 14 that under different salinity, the relative resistivity increases linearly with the gas phase fraction (The linear correlation coefficients are 0.997, 0.988 and 0.991 respectively), and the growth rates are similar, which are 0.553, 0.482 and 0.533 respectively. It can also be seen that with the increase of gas phase fraction, the fluctuation of relative resistivity is also greater, indicating that in the case of high gas phase fraction, the measurement error will be greater than that of low gas phase fraction. The reason for this phenomenon is that when the gas phase fraction of multiphase flow in the pipe is large, the spatial distribution uniformity of the phase fraction is low.

The experimental results show that even if the salinity of solution is different, the effect of gas phase fraction on relative resistivity is the same. In the following work, unless specified, 1.3% NaCl solution was used in the experiment.

(2)Effect of gas distribution on resistivity properties

According to the gas phase fraction, gas-water fluid is divided into bubble flow, slug flow, annular flow and so on. The resistivity properties of different flow patterns are very different. Sand production during hydrate exploitation usually occurs in the stage of large water production, and the liquid phase fraction is very large. At this time, the flow pattern of the pipeline fluid in the wellbore is bubble flow. So, in this work, we only studied the resistivity properties of the bubble flow. In the actual natural gas hydrates production wellbore, the bubble distribution in the pipeline fluid is not completely uniform. In this work, we studied the effect of uneven bubble distribution on the resistivity of gas-water fluid. Air stone, perforated circular plate with 3 mm holes and gas pipe with 8 mm diameter were separately used to inject gas into the loop to simulate different gas phase distributions. Figure 15 is the physical picture of air stone and perforated circular plate.

For the mixture composed of conductive phase and non-conductive phase, many researchers have proposed theoretical models for the relationship between resistivity and phase fraction. In this work, the model proposed by Maxwell [46] was used to compare with the experimental results.

It can be seen from Figure 16 that in the case of using air stone to inject gas, the relative resistivity is larger than that of using a gas pipe and a perforated plate to inject gas at the same gas phase fraction. The pores of air stone are small and dense, which can produce small and uniform bubbles. The bubbles produced by perforated plate and gas pipe are large and uneven. So, when the gas phase distribution is uniform, the relative resistivity of gas-water fluid is larger. The results show that even if the gas phase fraction is the same, the spatial distribution of the gas phase of the multiphase fluid will affect the relative resistivity. The more uniform the gas phase distribution, the greater the relative resistivity. In the design of measuring device, multiple pairs of electrodes can be used to avoid the effect of bubble distribution on relative resistivity which need to be further studied. In the following work, unless specified, gas was injected by air stone.

#### 3.2.2. Resistivity Properties of Water-Sand Fluid

(1)Effect of Sand Phase Fraction on Resistivity Properties

Some 60 μm quartz sand were injected into the flow loop to study the effect of sand phase fraction on relative resistivity. It can be seen from Figure 17 that there is a strong linear relationship between relative resistivity and sand phase fraction (The linear correlation coefficient is 0.982), and the growth rate of relative resistivity is 1.232. Combined with the previous results in this work, it can be seen that under the same phase fraction, the relative resistivity of water-sand fluid is greater than that of gas-water fluid. The reason is that sand particles are more evenly distributed in water than bubbles.

However, the relative resistivity of water-sand fluid is still lower than that calculated by the theoretical model. There are two reasons for it: (1) Although the sand particles are very small, they don’t meet the assumption of two-phase uniform distribution in the theoretical model. (2) When the measuring system is excited by high frequency, the distributed capacitance of the lead and other capacitance will lead to electric leakage, and then reduce the measured relative resistivity.

(2)Effect of Sand Particle Size on Resistivity Properties

In the actual hydrate exploitation process, the geological characteristics of hydrate reservoir, exploitation methods and sand production control measures will affect the particle size of sand produced from gas hydrate formation. In this work, the effect of sand particle size on the resistivity of water-sand fluid was studied by adding some quartz sand with different particle size and simulated seawater (26.5 g/L NaCl and 3.3 g/L MgSO_4_) into the measurement setup for static mixture.

It can be seen from Figure 18 that the effect of particle size on relative resistivity is almost the same when the sand phase fraction is different. In the range of 2 μm to 10 μm, the relative resistivity of water-sand mixture gradually increases with the particle size; when the particle size is 60 μm, the relative resistivity of water-sand mixture is less than the relative resistivity of mixture with 10 μm sand.

The reasons for this phenomenon are as follows. On the one hand, when the sand particles are relatively small, the electric charges carried by the sand particles are relatively large, which decreases the electrical resistivity of the water-sand fluid. So, when the sand particle size is 2 μm to 10 μm, the larger the particle size is, the greater the relative resistivity is. On the other hand, when the sand particles are relatively small, the sand particles are easier to distribute uniformly, and the more uniform the distribution of sand particles, the higher the resistivity of water-sand fluid. So, when the sand particle size is 60 μm, the relative resistivity of mixture is less than the relative resistivity of the mixture with sand particle size 10 μm.

The results show that the sand particle size will affect the relative resistivity of water-sand fluid. In the actual natural gas hydrates exploitation, it is necessary to determine the distribution range of sand particle size in order to realize the accurate measurement of sand phase fraction.

#### 3.2.3. Resistivity Properties of Gas-Water-Sand Fluid

The resistivity characteristics of gas-water fluid and water-sand fluid have been discussed. However, in the actual natural hydrate production process, the pipeline often contains gas and sand particles at the same time, so the effect of sand particles on the resistivity properties of gas-water-sand fluid was studied in this work.

It can be seen from Figure 19 that the relative resistivity of gas-water-sand fluid with sand phase fraction of 0.0323 is roughly equal to that of water-sand fluid under the condition of the same gas phase fraction. It should be noted that the relative resistivity of gas-water-sand fluid is calculated based on the resistivity of water-sand fluid. The results show that the water-sand fluid can be regarded as a special fluid when the particle size of sand particles is small, and the sand particles will not affect the measurement of gas phase fraction.

### 3.3. Combined Resistivity Method

In order to avoid the effect of bubbles on the measurement process, an inline pipe gas-water separator was designed in this work, and its structure is shown in Figure 20a. The main body of the gas-water separator is a punched cylinder, the diameter of the holes is 2 mm, and the spacing of the holes is 1 mm. The separation principle of the gas-water separator is that when the gas-water-sand fluid enters the perforated cylinder, the liquid flows out from the holes, and the gas are concentrated inside the punched cylinder, thereby achieving the purpose of making the surface of the pipe wall as free of bubbles as possible.

In order to verify the effect of the measuring device composed of three kinds of electrodes and gas-water separator, we carried out experimental research in the flow loop. 60 μm Quartz sand was injected to the flow loop. The three electrodes adopt the cyclic measurement mode, that is, only one electrode measured at each time.

#### 3.3.1. Measurement Effect of the Opposite-Wall Electrodes

The opposite-wall electrodes were installed at 200 mm upstream of the gas-water separator in this work. It can be seen from Figure 21 that when the gas fraction is greater than 0.087, the relative resistivity of multiphase fluid doesn’t increase and far less than the relative resistivity of multiphase fluid without gas-water separator.

The reason is that the existence of the gas-water separator leads to the concentration of bubbles in the center of the pipe. According to the previous results in this work, the more concentrated in the center of the pipe the bubbles are, the smaller the relative resistivity of gas-water fluid is. In actual applications, the position of the opposite-wall electrodes needs to have a certain distance from the gas-water separator to avoid effect of the gas-water separator on the opposite-wall electrodes. The value of distance needs to be further studied.

#### 3.3.2. Measurement Effect of the Side-Wall Electrode

It can be seen from Figure 22 that the relative resistivity measured with the side-wall electrode increases linearly (Linear correlation coefficient R = 0.941) with the gas phase fraction without gas-water separator; the relative resistivity measured with the side-wall electrode is almost unchanged with gas-water separator. It can be seen from Figure 23 that the relative resistivity measured by the side-wall electrode increases linearly (Linear correlation coefficient R = 0.888) with the sand phase fraction.

The results show that the gas-water separator designed in this work has a good effect and can avoid the effect of bubbles on the side wall electrodes. The side-wall electrode can measure the resistivity of water-sand fluid in gas-water-sand fluid.

#### 3.3.3. Measurement Effect of the Micro Electrode

It can be seen from Figure 24 and Figure 25 that the relative resistivity measured by the micro electrode is almost unchanged with gas phase fraction and sand phase fraction. The results show that the micro electrode can measure the resistivity of pure solution in gas-water-sand fluid.

## 4. Conclusions

In this work, the electrical resistivity properties of different gas-water-sand fluid were experimentally investigated using the multiphase flow loop experiment and static solution experiment. The effect of gas phase fraction and gas bubbles distribution, sand fraction and sand particle size on the relative resistivity of the multiphase fluid were systematically studied. A novel combined resistivity method was developed, and its measuring effects for the phase fraction of gas-water-sand fluid in wellbore during hydrate reservoirs production were studied. The operating parameters and the measurement devices were also optimized. The following conclusions are obtained:
(1)The relative resistivity of the gas containing fluid increases linearly with the increase of gas phase fraction in both gas-water and gas-water-sand fluid. And it is independent on the salinity, but decreases with the non-uniform spatial distribution and large bubble size of gas phase. For the gas supply of bubble stone, when the gas phase fraction is lower than 20%, the relationship between relative resistivity and gas phase fraction meets the following requirements: $\rho_{r,G}=1+0{.523\alpha }_{G}$.The presence of sand particles doesn’t have obvious effect on the relationship between relative resistivity and gas phase fraction in gas-water-sand fluid. For the water-sand fluid, the relative resistivity increases linearly with the increase of sand phase fraction, and is affected by sand particle diameter. For water-sand fluid containing 60 μm quartz sand, when sand phase fraction is lower than 7%, the relationship between relative resistivity and sand phase fraction meets the following requirements: $\rho_{r,S}=1+1{.637\alpha }_{S}$.(2)The electrical resistivity of the pipe wall material has a great influence on the electrical measurement. The inner wall of the measurement pipe section should be insulated to avoid wall leakage. The electrode shape doesn’t affect the relative resistivity measurement. The sensitive field of the opposite-wall electrodes with 4 mm diameter is concentrated on a flat cross-section with little axial extension in gas-water-sand fluid. The optimized excitation frequency is about 10 kHz.(3)The combined resistivity method employs three kinds of electrodes and an in-line gas-water separator, can effectively measure the sand fraction of the gas-water-sand fluids by a micro electrode and a side-wall electrode. The gas fraction could be measured accurately by using multiple pairs of opposite-wall electrodes to resolve the effect of gas non-uniform distribution. It will have a good application potential in marine natural gas hydrates exploitation.


## Figures and Tables

**Figure 1 entropy-24-00624-f001:**
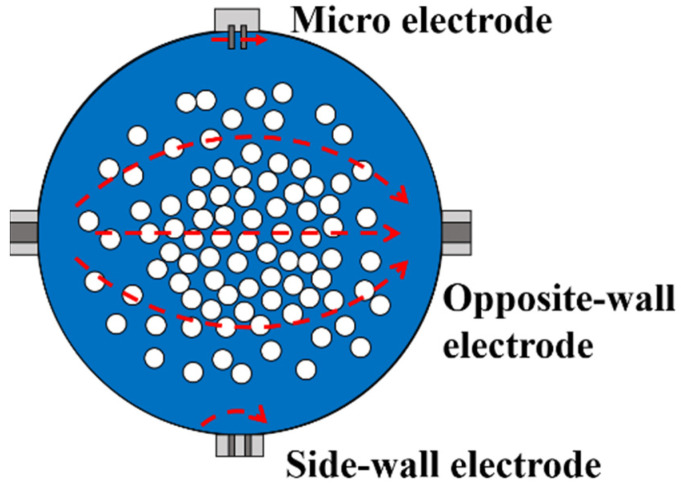
Schematic diagram of measurement principle of combined resistivity method.

**Figure 2 entropy-24-00624-f002:**
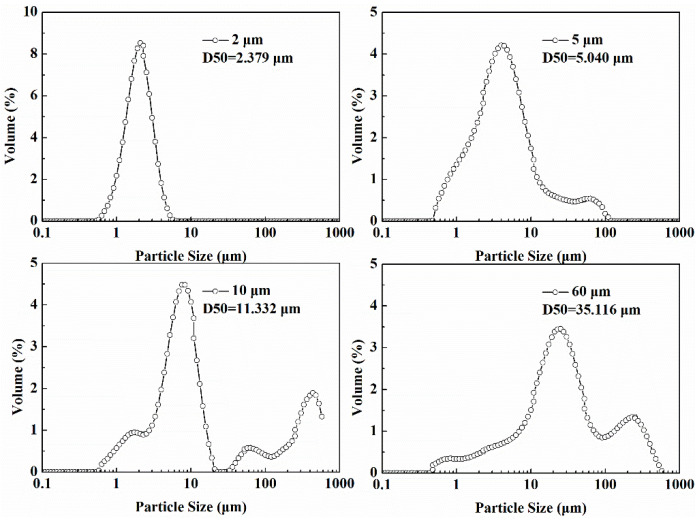
Particle size distribution of quartz sand.

**Figure 3 entropy-24-00624-f003:**
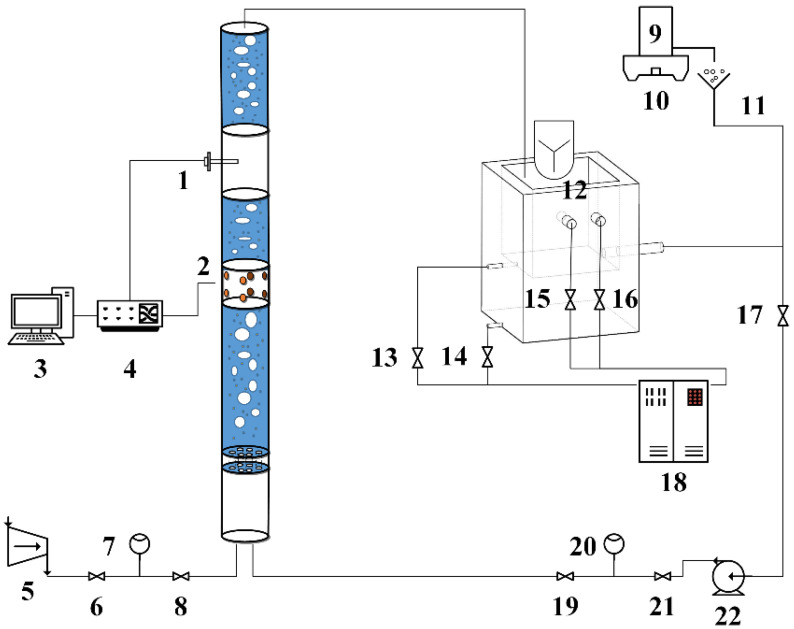
Schematic diagram of the multiphase flow loop. (1—Thermocouple; 2—Electrodes; 3—Computer; 4—Agilent 34970A; 5—Air compressor; 7, 20—Flow meter; 6, 8, 13, 14, 15, 16, 17, 19, 21—Valve; 9—Sand storage tank; 10—Electronic balance; 11—Sand injection funnel; 12—Mechanical stirring; 18—Water bath; 22—Water pump).

**Figure 4 entropy-24-00624-f004:**
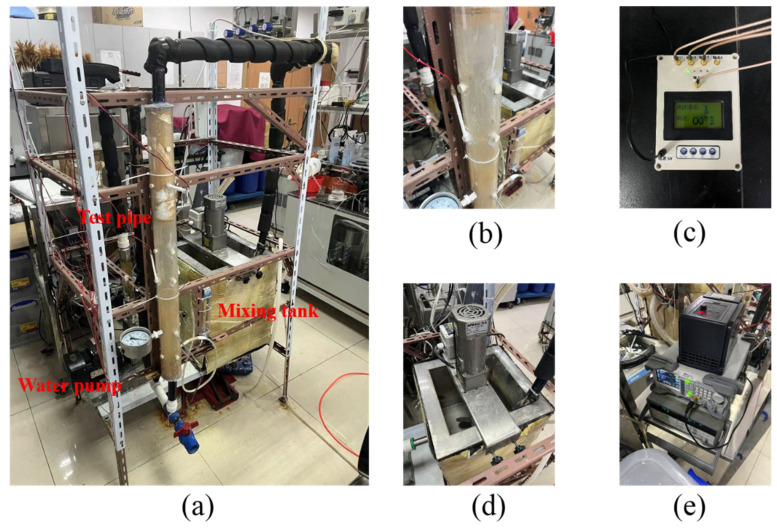
Device photos of the multiphase flow loop. (**a**) Experimental loop; (**b**) Test pipe section; (**c**) Switching circuit; (**d**) Mixing tank; (**e**) Signal generator, Agilent 34970A and the frequency inverter.

**Figure 5 entropy-24-00624-f005:**
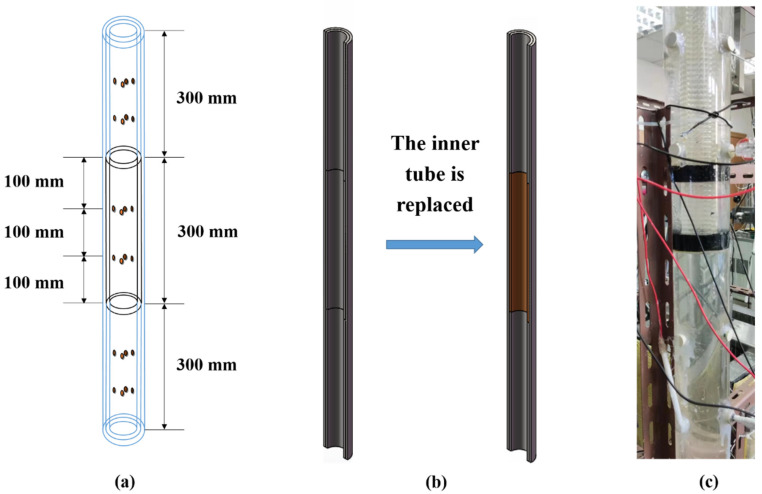
(**a**) Structure of test pipe section; (**b**) Replacement process of inner pipe; (**c**) Physical drawing of test pipe section.

**Figure 6 entropy-24-00624-f006:**
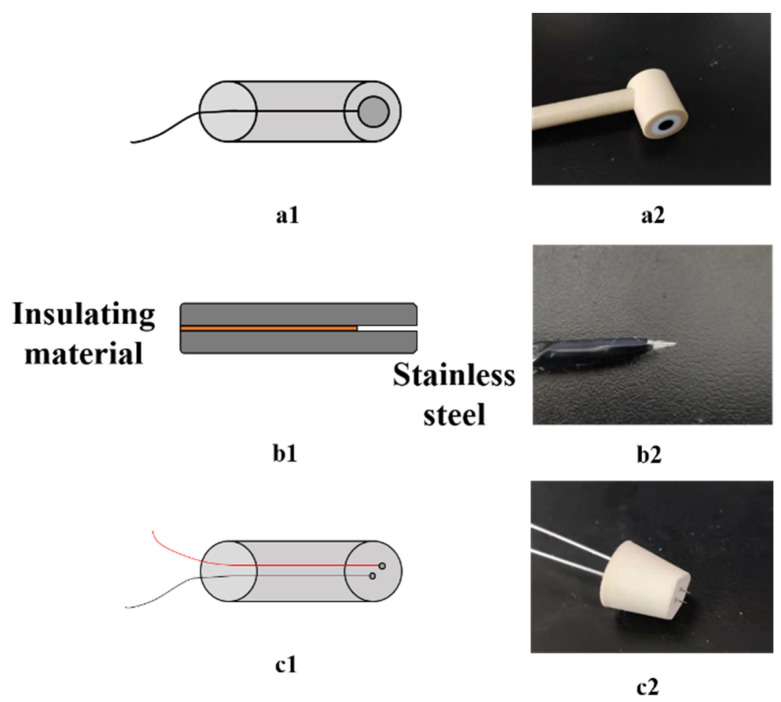
(**a1**) Schematic diagram of the opposite-wall electrode; (**a2**) The opposite-wall electrode; (**b1**) Schematic diagram of the micro electrode; (**b2**) The micro electrode; (**c1**) Schematic diagram of the sidewall electrode; (**c2**) The side-wall electrode.

**Figure 7 entropy-24-00624-f007:**
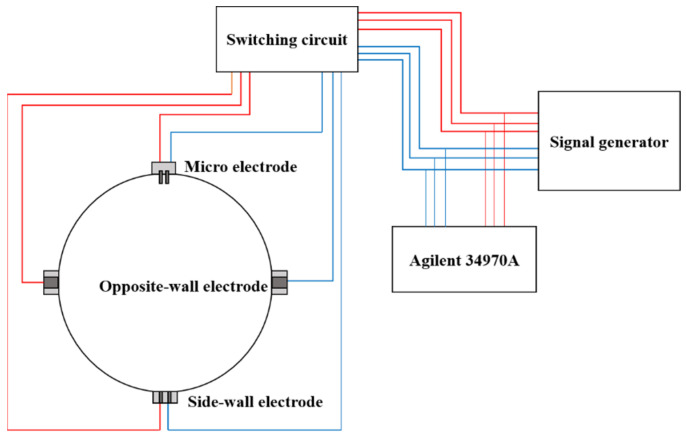
Diagram of resistance test circuit.

**Figure 8 entropy-24-00624-f008:**
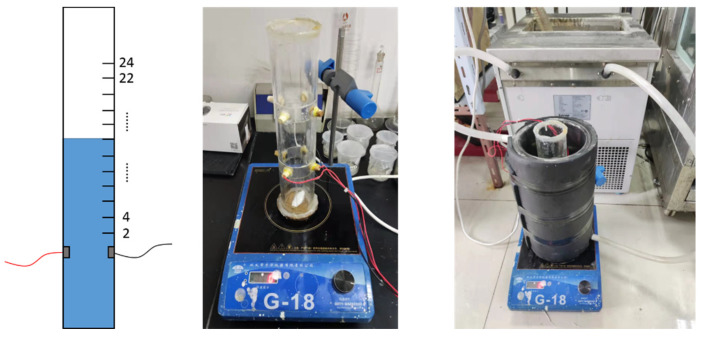
Schematic and physical diagram of static device.

**Figure 9 entropy-24-00624-f009:**
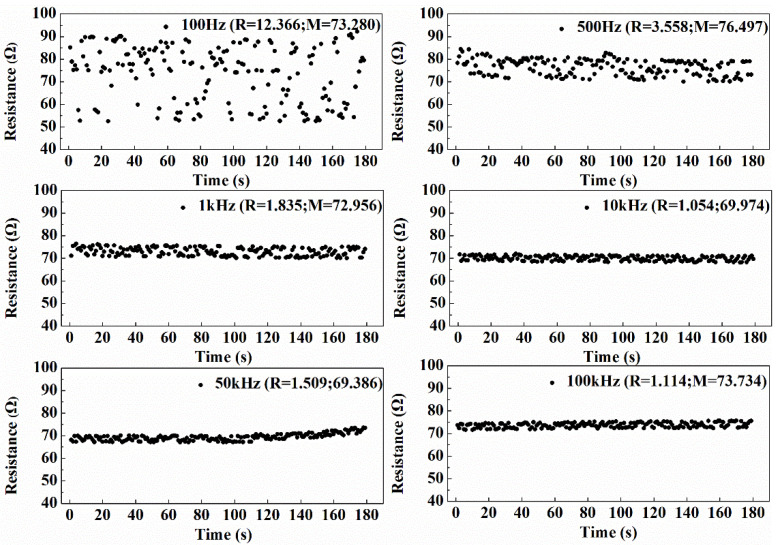
Resistance measured with different excitation frequencies. (R is the standard deviation and M is the mean value).

**Figure 10 entropy-24-00624-f010:**
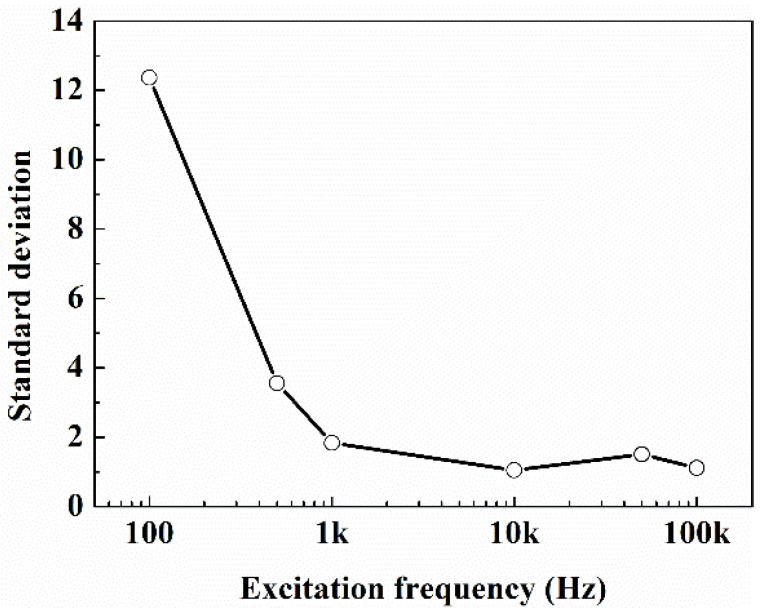
Standard deviation of resistance at different excitation frequencies.

**Figure 11 entropy-24-00624-f011:**
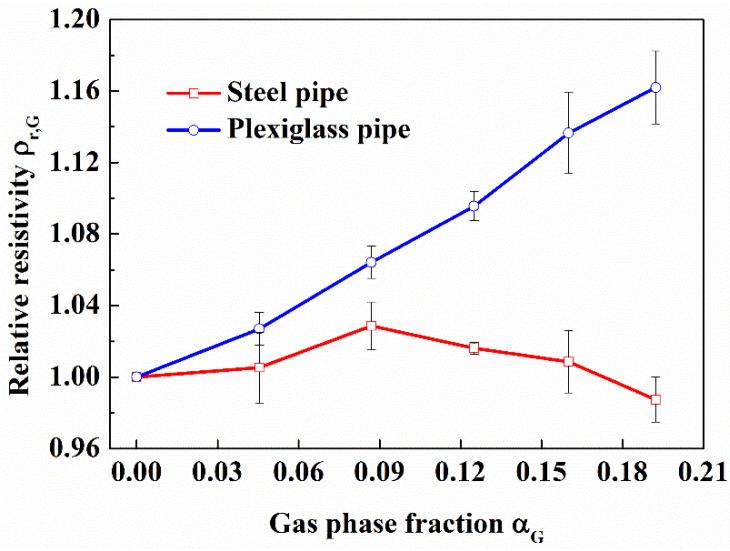
Comparison of the effect of pipelines materials on electrical resistance measurement under different gas phase fraction. (In the flow loop).

**Figure 12 entropy-24-00624-f012:**
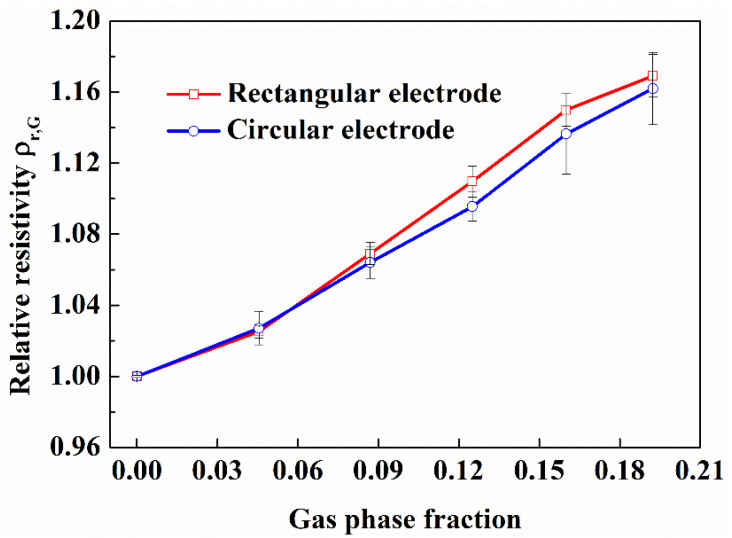
Comparison of electrode shape on electrical resistance measurement under different gas phase fraction in flow loop experiment.

**Figure 13 entropy-24-00624-f013:**
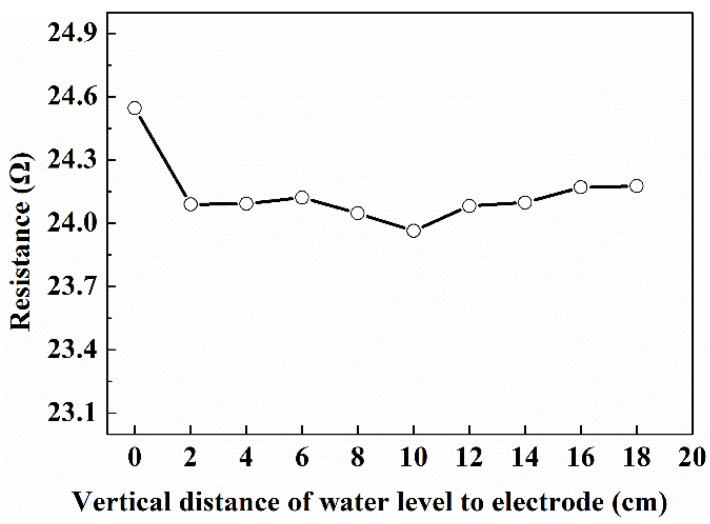
The relationship between the resistance value measured by the opposite-wall electrodes and the distance between the upper surface of the solution in the cylinder and the upper end of the electrode. (1.3% NaCl solution; in the static experiment setup).

**Figure 14 entropy-24-00624-f014:**
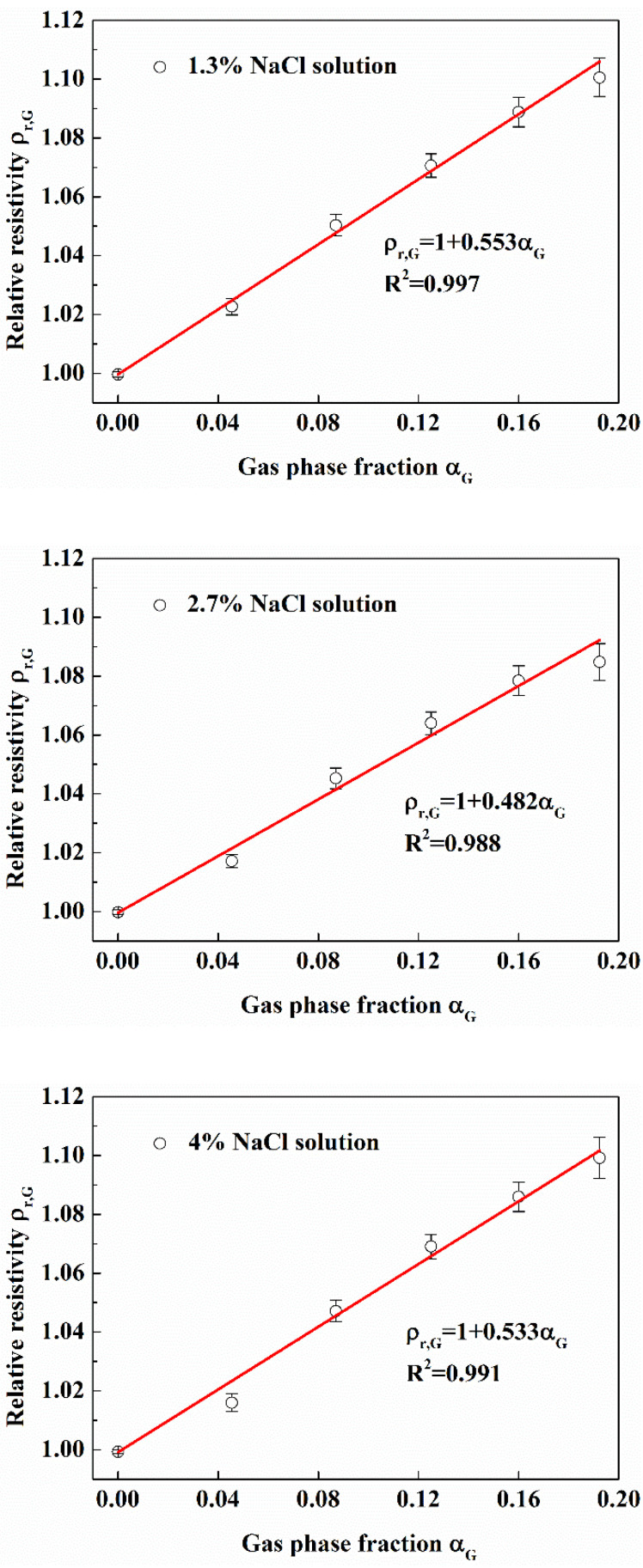
Effect of gas phase fraction on relative resistivity under different salinity conditions. (Sand phase fraction is 0; gas was injected by air stone; in the flow loop).

**Figure 15 entropy-24-00624-f015:**
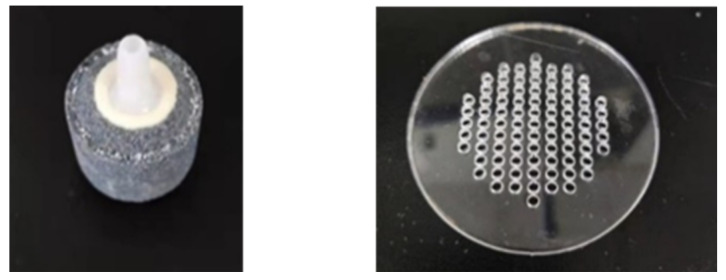
The air stone and the perforated circular plate.

**Figure 16 entropy-24-00624-f016:**
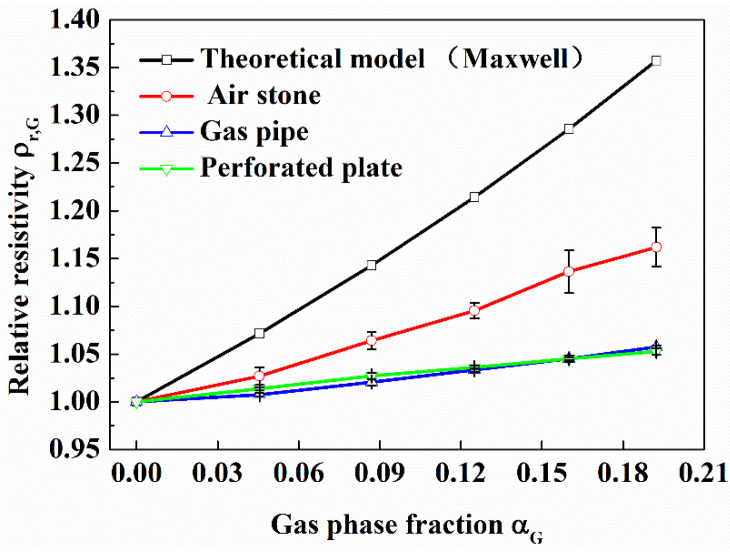
Effect of gas phase distribution on relative resistivity. (Sand phase fraction is 0; in the flow loop).

**Figure 17 entropy-24-00624-f017:**
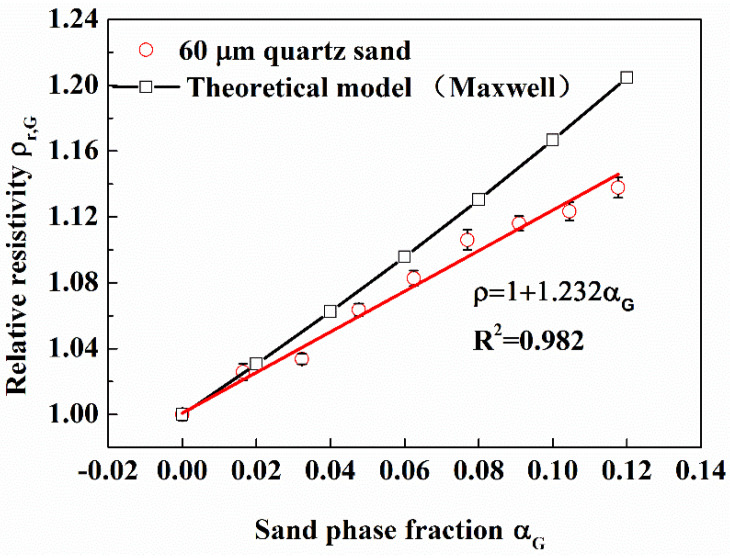
Effect of sand phase fraction on relative resistivity. (60 μm quartz sand were injected; in the flow loop).

**Figure 18 entropy-24-00624-f018:**
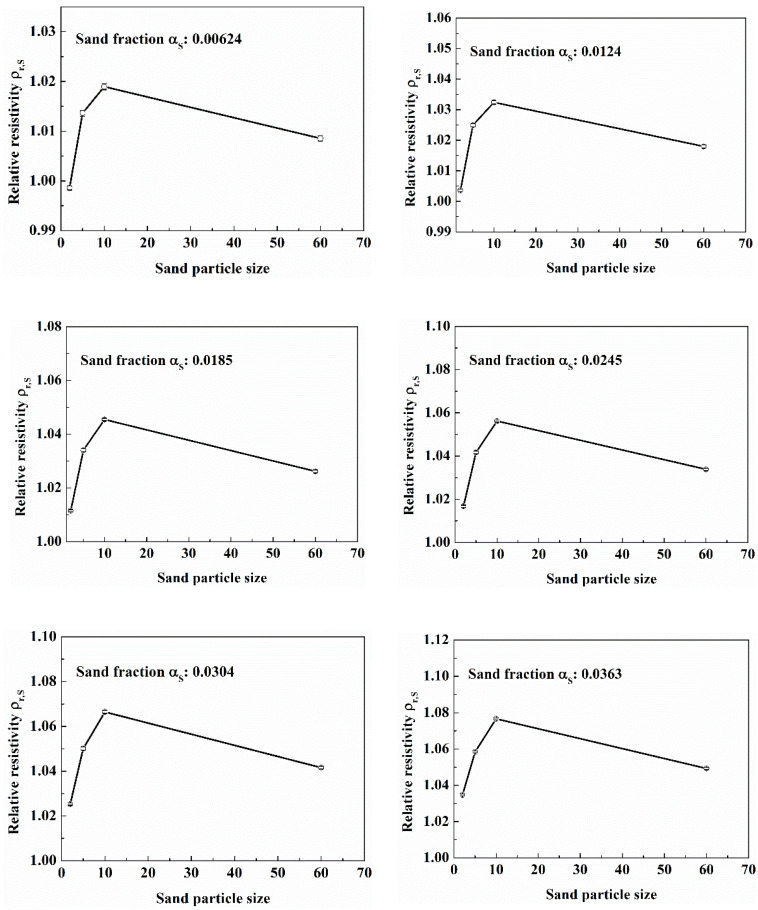
Effect of sand particle size on relative resistivity with different sand phase fraction. (Simulated seawater; in the static experiment setup).

**Figure 19 entropy-24-00624-f019:**
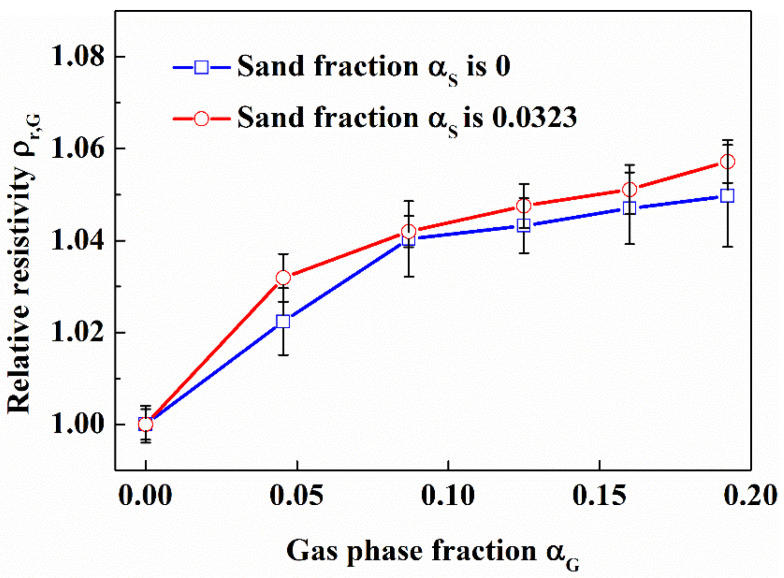
Effect of sand phase fraction on the relative resistivity of gas-water-sand fluid. (60 μm quartz sand was injected; in the flow loop).

**Figure 20 entropy-24-00624-f020:**
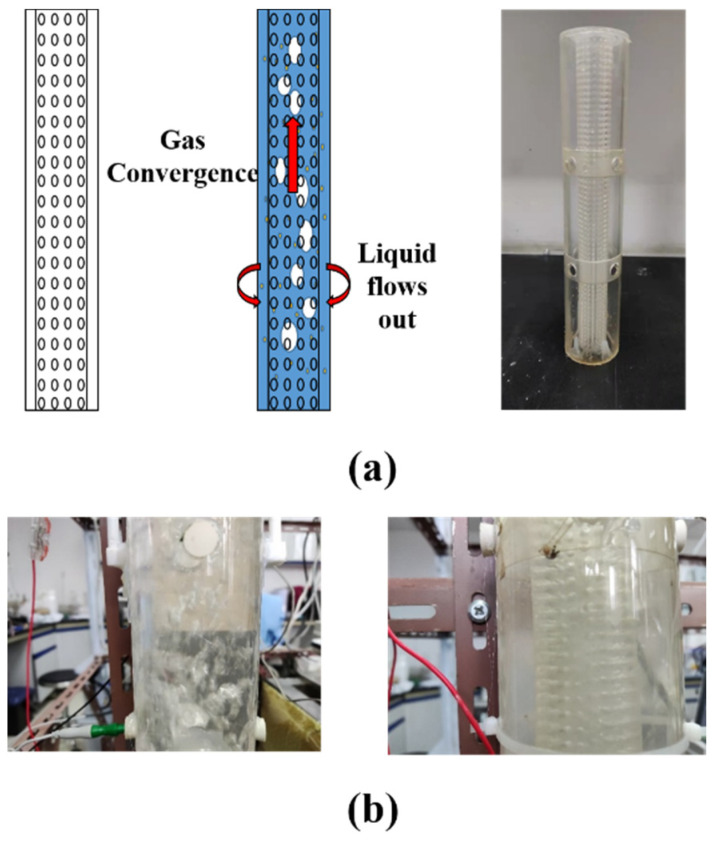
(**a**) Schematic and physical diagram of gas-water separator. (**b**) Effect of gas-water separator in the flow loop. (Gas phase fraction is 0.087; gas-water separator is used on the right).

**Figure 21 entropy-24-00624-f021:**
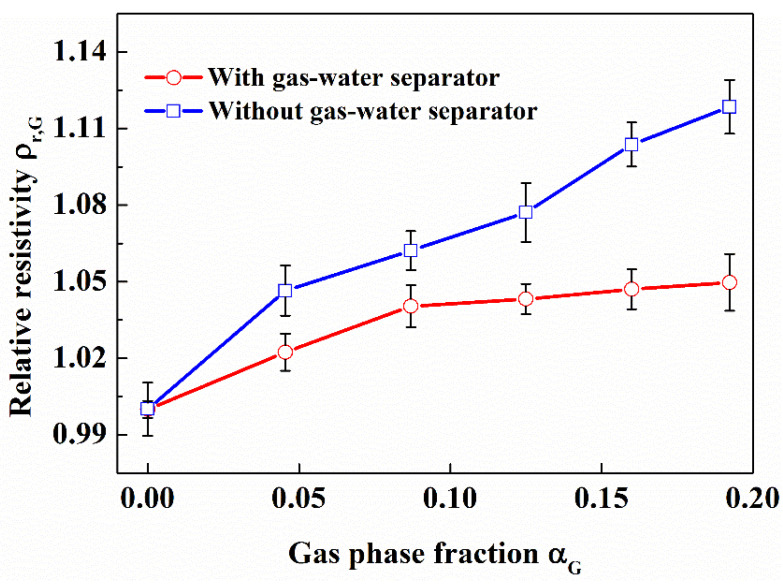
Effect of gas on opposite-wall electrodes with gas-water separator. (Sand phase fraction is 0.0323, in the flow loop).

**Figure 22 entropy-24-00624-f022:**
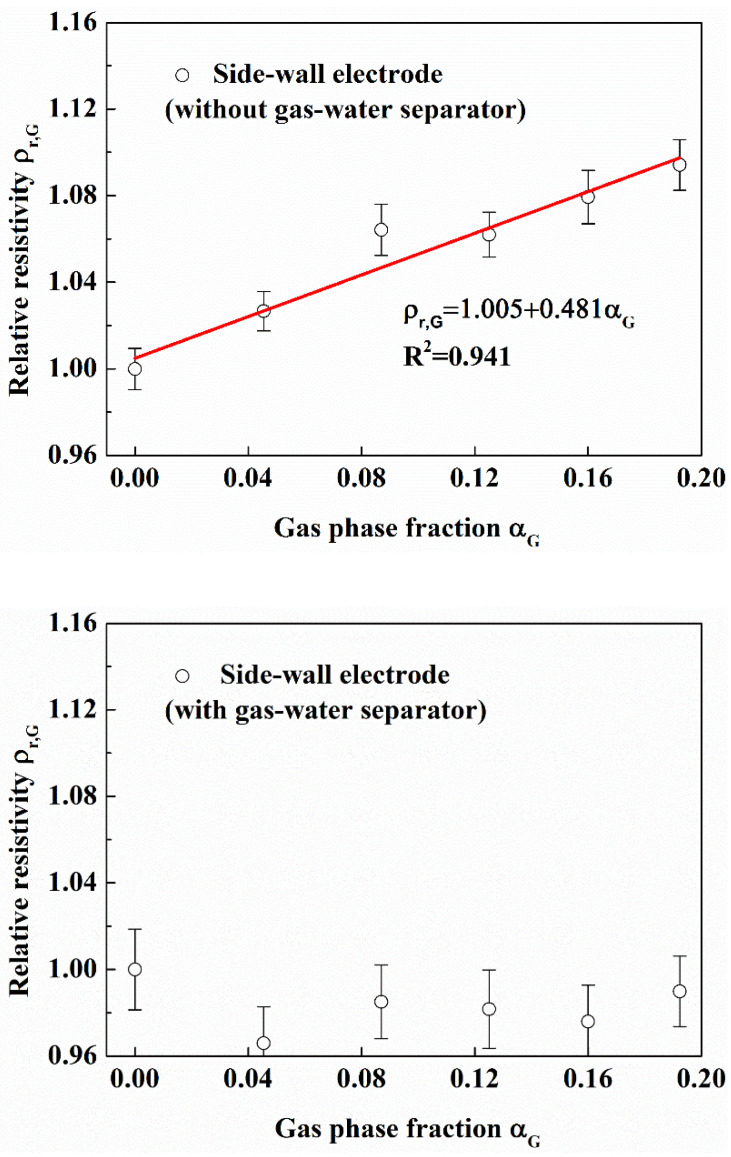
Comparison of effect of gas on side-wall electrode without or with gas-water separator. (Sand phase fraction is 0.0323, in the flow loop).

**Figure 23 entropy-24-00624-f023:**
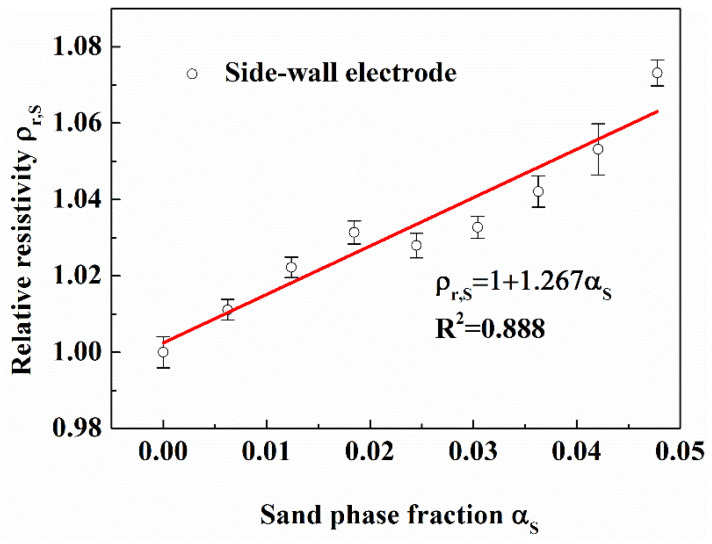
Effect of sand particles on side-wall electrode. (Gas phase fraction is 0.087, in the flow loop).

**Figure 24 entropy-24-00624-f024:**
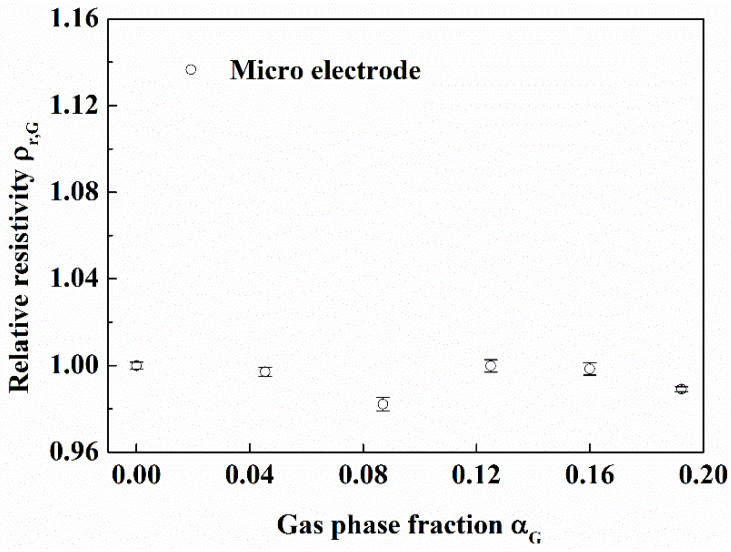
Effect of gas on micro electrode. (Sand phase fraction is 0.0323, in the flow loop).

**Figure 25 entropy-24-00624-f025:**
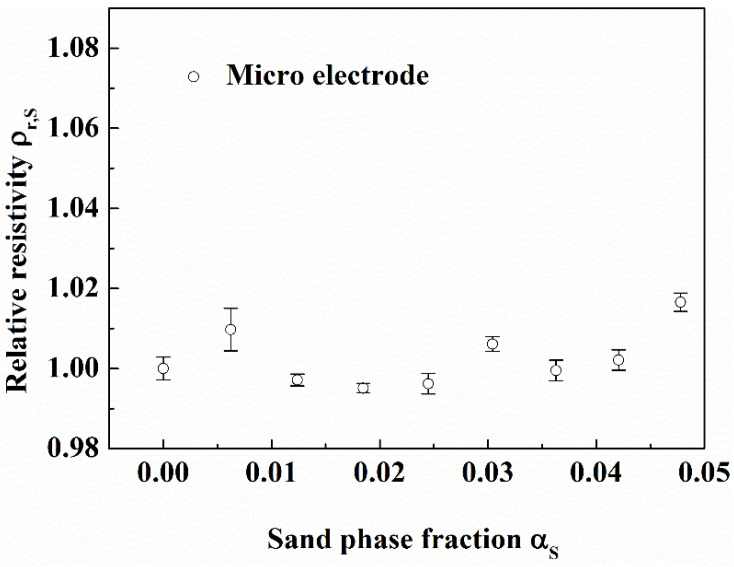
Effect of sand particles on micro electrode. (Gas phase fraction is 0.087, in the flow loop).

**Table 1 entropy-24-00624-t001:** Comparison of different techniques of electrical conductivity method.

Method	Advantage	Disadvantage	Object	Result
Erosion method [21,22]	Can measure the solid phase fraction in non-conductive fluid	The sheet metal needs to be replaced frequently	Oil-sand fluid	Sand phase fraction
Wire mesh sensor method [23,24,25,26]	The imaging algorithm is simple	It interferes with the fluid and the wire mesh is easy to be worn and damaged	Gas-water fluid	Gas phase distribution
Electrical tomography method [27,28,29,30,31]	Without disturbing the fluid flow, the spatial distribution information of phase fraction can be obtained.	The imaging algorithm is complex, requires high computational cost, and the imaging accuracy is low.	Gas-water fluid	Gas phase distribution
Conductance probe method [32]	It can be combined to measure the spatial distribution of phase fraction	It interferes with the fluid and the measurement range is very small	Gas-water fluid	Local gas phase fraction
Capacitively coupled contactless conductivity detection method [33]	The electrode is not in direct contact with the fluid, so it will not be corroded	Complete insulation of pipe wall material is required	Gas-water fluid	Gas phase fraction
Contacting electrical method [36]	The structure is simple and can be combined freely	The electrode is easily corroded	Gas-water fluid	Gas phase fraction

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
