# Peer review of "Investigation of Gas-Water-Sand Fluid Resistivity Property as Potential Application for Marine Gas Hydrate Production"

_entropy, 2022, doi:10.3390/e24050624_

Round 1

Reviewer 1 Report

The authors propose a new method for phase fraction measurment. The method is sound however, the paper layout needs to be changed significantly. At the moment the new method description appears towards the end. The method section should be moved to the first explaining the novelties in comparison to current state of art resistivity methods. A few more comments are below:

1. Authors have conducted a good literature review. However, while mentioning the different techniques of electrical conductivity method by different authors, it would also be useful to shed some light on the efficacy of these different methods which could later be used to compare with the method authors have proposed. 
2. More clear and good quality photo of the setup is needed with labels. 
3. Fig.4 is not clear and needs more clear discussion. The holes should be marked and their distances should be dimensioned in the figure. There are 5 pictures in Fig. 4 and should be listed as (a), (b) etc with captions for better understanding. 
4. I think it is better to discuss the underlying method first and then go into the discussion of the experimental setup. The current format discusses the experimental setup and electrodes and the reader is wondering why the electrodes have certain design etc. If there is a method section first, this would help reader understand the experimental setup much better. 
5. Authors must describe the working principle first of the electrical sensor. 
6. The symbols of rho_r,G is different in text and equations. 
7. It is not clear from sec 2.3.3 how the phase fraction is calculated from the resistivity measurements. This needs to be clearly demonstrated. 
8. Please describe what is polarisation effect of electrode and its consquences.
9. In sec.3.1.3 how was the gas fraction measured for the calibration process?

Reviewer 2 Report

The presented work will be interesting a wide range of readers and is a complex scientific work on the study of the electrical resistivity properties of the gas-water-sand fluid.
Firstly I think the title of the article is too long, it should be shortened.
For example 'Investigation of gas-water-sand mixture properties in multiphase fluid as application for marine gas hydrate production'.

It is also necessary to change the some drawings as they are overloaded information. For example, in Figure 10, there are 3 experimental curves for each material (steel pipe and plexiglass pipe).
If the experimental conditions are the same, then you just need to average the data for 3 experimental curves , indicating the measurement error.
The situation is similar with other drawings (Figures 11, 15, 17).
And last, since the work is clearly applied , it is necessary to indicate specific experimental data in the conclusions.

Round 2

Reviewer 1 Report

Paper can be accepted